# The role of atomic interactions in cavity-induced continuous time crystals

**Christian Høj Johansen[1], Johannes Lang[1,2] and Francesco Piazza[1,3]**

**1** Max-Planck-Institut für Physik komplexer Systeme, 01187 Dresden, Germany
**2** Institut für Theoretische Physik, Universität zu Köln,
Zülpicher Straße 77, 50937 Cologne, Germany
**3** Theoretical Physics III, Center for Electronic Correlations and Magnetism,
Institute of Physics, University of Augsburg, 86135 Augsburg, Germany

⋆ francesco.piazza@uni-a.de

## Abstract

We consider continuous time-crystalline phases in dissipative many-body systems of atoms in cavities, focusing on the role of short-range interatomic interactions. First, we show that the latter can alter the nature of the time crystal by changing the type of the underlying critical bifurcation. Second, we characterize the heating mechanism and dynamics resulting from the short-range interactions and demonstrate that they make the time crystal inherently metastable. We argue this is generic for the broader class of dissipative time crystals in atom-cavity systems whenever the cavity loss rate is comparable to the atomic recoil energy. We observe that such a scenario for heating has several similarities to the one proposed for preheating in the early universe, where the oscillating coherent inflation field decays into a cascade of exponentially growing fluctuations. By extending approaches for dissipative dynamical systems to our many-body problem, we obtain analytical predictions for the parameters describing the phase transition and the heating rate inside the time-crystalline phase. We underpin and extend the analytical predictions of the heating rates with numerical simulations, which also show that the metastable regime exists when the inherent stochastic nature is taken into account.



# 1   Introduction

Following the first conceptualization of time-crystalline phases of matter [1,2], it was quickly proven that such phases cannot appear in thermal equilibrium [3–5]. However, it is possible to realize such phases in periodically driven systems, both closed [6–11] and dissipative [12,13].

Among the latter, systems of atoms in optical cavities have emerged as an ideal platform to realize continuous time-crystalline phases [14–16], where an effectively time-independent drive of the atomic system is counterbalanced by the loss of photons out of the cavity mirrors. In these phases, continuous time-translation invariance is spontaneously broken, and oscillations persist even though the system possesses a macroscopic number of degrees of freedom, among which energy can be redistributed via interactions.

Since the phase space of scattering by cavity-mediated interactions between atoms is limited due to their long range, redistribution of energy through these processes is inefficient [17–19]. However, the intrinsic atomic short-range interactions allow for efficient redistribution of energy among the atoms. Indeed, experiments show strong indications that these interactions are one of the main fundamental limiting factors to the measured lifetime of the time crystal [12].

Despite their crucial role, short-range atomic interactions have not been theoretically investigated so far in a systematic way for continuous time crystals in atom-cavity setups. In this work, we undertake this task. We not only provide a complete picture of the possible destabilization processes but also show that short-range interactions can alter the nature of the time crystal itself.

In Section 2 we describe the model under consideration, which has a simple and experimentally realizable mechanism for the appearance of time-crystalline phases for an interacting BEC coupled to two cavity modes [20]. In Section 3 approaches for classical non-linear dissipative systems is extended to our many-body problem and we obtain an analytic description of the time crystal in terms of cavity-induced critical bifurcations and show how interatomic interactions can modify the nature of the latter. These results are used in Section 4, where we identify the dominating scattering processes responsible for energy redistribution among the atoms. Using these scattering processes we compute the dependence of the energy-redistribution rates on external parameters identify the corresponding time scales. Lastly, in Section 5, we show numerically that for realistic noise levels, the metastability of the time crystal is fully determined by the redistribution rates due to inter-atomic interactions.

The analytic understanding of the results, underpinned with numerical analysis, allows for a deep insight into the generic features of the phenomenology beyond the specific model considered and provides orientation for future investigations both in theory and experiment.

## 2 Model

The system considered is an ultracold gas of bosonic atoms in a BEC state, dispersively coupled with equal strength to two modes of an optical cavity. In this regime, a photon imparts a recoil momentum of $Q = 2\pi/\lambda$ to an atom, with $\lambda$ being the wavelength of the photon in a given mode. In the thermodynamic limit, the atomic BEC at momentum $k$ is described by a complex field $\psi_k$ satisfying the Gross-Pitaevski mean-field equations. Furthermore, in the limit of a small transverse extend of the BEC compared to the cavity waist we can simplify the model to one spatial dimension [19,20]

$$i\partial_t \psi_k = k^2 \psi_k + U \sum_{q,q'} \psi_q \psi_{q'} \bar{\psi}_{q+q'-k} + \frac{\tilde{\eta}}{\sqrt{2}} \sum_{j=1,2} \text{Re}(\phi_j)(\psi_{k+Q} + \psi_{k-Q}), \tag{1}$$

where the bar denotes complex conjugation. This equation has been written in units of the recoil energy $E_R = \hbar^2 Q^2/2m$ and in the rotating frame of the laser. The time-dependence of the fields is kept implicit and the atom field has been normalized to 1. The cavity-mode wavelengths have been chosen to be equal, as we assume the modes differ in the transverse direction [20]. The coupling strength $\tilde{\eta}$ can experimentally be tuned by the strength of the transverse pump while the atoms are interacting with each other through a contact interaction of strength $U$. The complex field $\phi_j$ corresponds to the coherent cavity-field amplitude which satisfies the equation

$$i\partial_t \phi_j = (\Delta_j - i\kappa)\phi_j + \frac{\tilde{\eta}}{2\sqrt{2}} \sum_{k=-\infty}^{\infty} \bar{\psi}_k (\psi_{k+Q} + \psi_{k-Q}), \tag{2}$$

where the cavity field has been normalized by the square root of the atom number. The cavity linewidths, $\kappa$, have been assumed to be identical for both modes. In the following we will consider $\kappa$ on an energy scale similar to the recoil energy, as realized for instance in [21]. In the actual implementation of the dispersive atom-cavity coupling, the characteristic frequency of each cavity mode $\Delta_j$ corresponds to the detuning of the mode frequency with respect to laser-driven two-photon transitions [20]. The steady state of this model can break time-translation invariance when the two detunings have opposite signs. With this in mind the detunings are parametrized as $\Delta_1 = -(\Delta - \frac{\delta}{2})$ and $\Delta_2 = \Delta + \frac{\delta}{2}$. By choosing $0 < \delta < 2\Delta$ the negative detuning has the smallest amplitude $|\Delta_1| < |\Delta_2|$.

The mean-field description of this system becomes exact in the thermodynamic limit [22]. It, however, does not give rise to a unique steady state of the atomic system. To determine the latter, one needs to include quantum fluctuations [23]. The time scale needed to reach this steady state does grow inversely with the system size. For the large atomic clouds considered here, the relaxation to this steady state is thus irrelevant on the experimental time scale. Additionally, for a system with a finite number of atoms, the openness of the cavity gives rise to an additional stochastic term in the cavity equation. The strength of the stochastic term scales with $\kappa/\sqrt{N}$, with N being the number of atoms. We will initially consider the limit of $N \to \infty$ and in Sec. 5 show that the conclusions remain valid also for finite system sizes.

## 3 Nature of the time crystal

Below a critical coupling strength $\eta_c$, all atoms are in the homogeneous state $\psi_0$, and the coherent part of the cavity fields is empty. This configuration is denoted as the normal phase (NP) and it is always a fixed point of the equations of motion Eqs. (1) and (2). As $\tilde{\eta}$ is increased beyond $\eta_c$ the NP fixed point becomes unstable and the system enters a state where

a fraction of the atom population is transferred to $\psi_{\pm Q}$ and the coherent fields of the cavity becomes finite. This symmetry-broken state is often referred to as the superradiant (SR) or self-organized state [24, 25]. The frequency $\omega_c$ of the excitation becoming undamped above $\eta_c$, can be derived through a linear expansion around the NP fixed point [26] (see [27] for an alternative approach). This is done by considering a small perturbation to the state NP state and only keeping the terms linear in the perturbation. One can then derive an equation for when the perturbation becomes unstable, indicated by a zero real part of the eigenvalue of the resulting Jacobian of the linear system.

The frequency of the unstable perturbation is determined by the imaginary part of the eigenvalue. It is found that an unstable mode can appear for three different frequencies of the critical mode. The determination of which the three possible modes that ends up manifesting in the system is set identifying which of the three modes require the smallest corresponding values of $\eta_c$. The first of the three potential instabilities is a static solution with $\omega_c = \omega_{c,s} = 0$. The second frequency at which the system can become critical is a resonance at the energy of the Bogoliubov excitation of the BEC at the recoil momentum with frequency $\omega_c = \omega_a = \sqrt{E_R(E_R + 2U)}$ and lastly, there is a possibility of an instability at a frequency given by

$$\omega_c = \sqrt{\frac{\delta^2}{4} + \sqrt{(4\Delta^2 - \delta^2)(\Delta^2 + \kappa^2)} - \Delta^2 - \kappa^2}, \tag{3}$$

which is solely determined by cavity parameters, that is, it does not depend on $U$ and $E_R$. This feature, which can be attributed to the fact that the cavity is the only dissipation channel, implies a robustness of this self-sustained periodic signal to perturbations of the nonlinear medium that causes this signal to appear in the first place. To determine which mode that becomes unstable one identifies which mode leads to the smallest real critical coupling given by

$$\eta_c = \lim_{\kappa_a \to 0} \sqrt{\frac{\omega_a^2 + \kappa^2 - \omega_c^2}{E_R \sum_{j=1,2} \frac{\Delta_j(\Delta_j^2 + \kappa^2 - \omega_c^2)}{\omega_c^4 + 2\omega_c^2(\kappa^2 - \Delta_j^2) + (\Delta_j^2 + \kappa^2)^2}}}, \tag{4}$$

where $\kappa_a$ is the atom lifetime which we, for simplicity consider infinitely long. always depends on both cavity and atom parameters [26]. The phase diagram will therefore depend on all parameters of the model.

In Fig. 1(a) the frequency of the critical mode at $\eta_c$ is plotted as a function of $\kappa$ and $\Delta$ and is a good order parameter for distinguishing the three different phases of the system. For $\Delta < \delta/2$ both cavity modes have a positive detuning and the system always exhibits static superradiance (SSR), characterized by a critical mode with zero frequency. SSR requires a finite critical atom-cavity coupling such that the critical mode is a polariton. For $\Delta > \delta/2$ one of the modes acquires a negative detuning. Differently from a positively-detuned mode, a negatively-detuned one disfavors a superradiant density modulation. The competition between the two cavity modes induces an oscillating superradiant phase (OSR) [20, 28], which also requires a finite coupling strength such that the critical mode is again a polariton. Instead, when the solution with $\omega_c = \omega_a$ has a real critical coupling, $\eta_c$ in Eq. (4) vanishes, making the critical mode purely atomic. We refer to this instability as the atomic instability (AI).

Both the OSR and AI critical modes break continuous time-translation invariance and can thus potentially signal a continuous time-crystal phase. However, whether the latter is stable is determined by non-linear effects not included so far. In order to capture these in the present interacting many-body system, we perform a systematic perturbative expansion in the relative distance from the critical point $\eta = (\tilde{\eta} - \eta_c)/\eta_c$. The resulting effective non-linear equation is of the Stuart-Landau form (see e.g. [29]), and is an equation of motion for the collective degrees of freedom which are excited in the SR phases. These degrees of freedom constitute

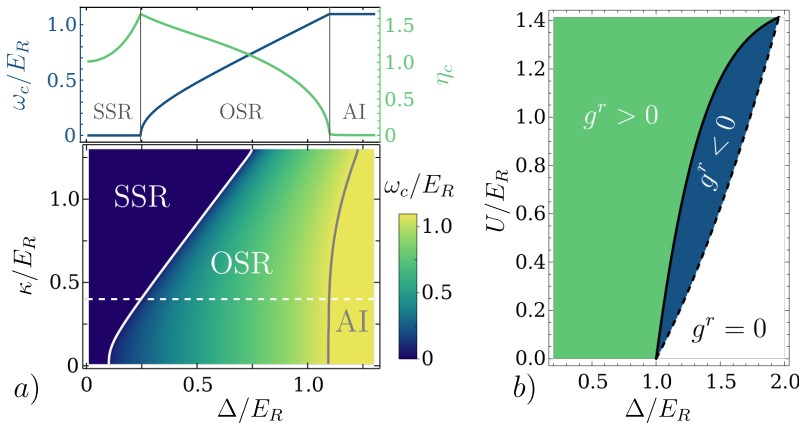

Figure 1: The critical frequency of the instability is shown in the lower plot of a) as a function of $\Delta$ and $\kappa$. By tuning $\Delta$ the critical mode change from exhibiting static to oscillating superradiance and a purely atomic instability over a large range of cavity loss rates. The upper plot shows the critical frequency and coupling along the dashed line in the lower plot. In b) the sign of the cubic interaction as a function of $\Delta$ and $U$ is plotted for $\kappa = 0.4E_R$. This determines the stability of the symmetry-broken state beyond the linear analysis. For for the entire figure $\delta = 0.2E_R$.

the so-called center manifold and are defined by the critical mode, which is composed of both cavity modes as well as of the zero and recoil momentum components of the BEC, $\psi_0$ and $\psi_{\pm Q}$. Within the center manifold and to leading order in $\eta$ the recoil momentum component is given by

$$\psi_{\pm Q}(t) = \sqrt{\eta}R\left(c_+ e^{i\omega_c t} + c_- e^{-i\omega_c t}\right), \tag{5}$$

with $c_\pm$ being the atomic components of the critical-mode eigenvector obtained from the linear analysis [26]. The cavity fields have the same form with $c_\pm$ replaced by the cavity components of the critical mode. Finally, since to leading order the only occupied atom components are $\psi_0$ and $\psi_{\pm Q}$, these are linked by normalization such that

$$\psi_0 = \sqrt{1 - \left|\psi_Q\right|^2 - \left|\psi_{-Q}\right|^2} \sim b_0 + b_+ e^{i2\omega_c t} + \bar{b}_+ e^{-i2\omega_c t}, \tag{6}$$

with $b_0 = 1 - \eta R^2\left(|c_+|^2 + |c_-|^2\right)$ and $b_+ = -\eta R^2 c_+ \bar{c}_-$. The perturbative approach yields an equation of motion for the amplitude, $R$, in the symmetry-broken phase:

$$\dot{R} = \gamma R - g^r R^3, \tag{7}$$

where $\gamma$ is the exponential growth rate of the critical mode obtained from the linear analysis, which in this case can be shown to be positive. The non-linearity of the center manifold or in other words, the strength of the self-interaction of the excitations present in the critical mode, is quantified by $g^r$ (see Appendix A.2 for closed expressions for these quantities and the proof of the sign of $\gamma$). For stable time-crystalline and static solutions, $R$ must be time-independent, real, and positive:

$$R = \sqrt{\frac{\gamma}{g^r}} > 0. \tag{8}$$

As $\gamma > 0$, our analytic solutions can only be stable if $g^r > 0$. This is physically clear since otherwise the attractive self-interaction would lead to a first order transition into a phase that requires higher-order non-linearities for stabilization.

The sign of $g^r$ is shown in Fig. 1(b). If $\omega_c$ is pushed to $\omega_a$, $g_r = 0$ i.e. the self-interaction vanishes as the critical mode is purely atomic, which corresponds to the white region in Fig. 1(b). As the fraction $\gamma/g^r$ goes to zero as $\omega_c$ approaches $\omega_a$ (see Eq. (A.23)), the AI phase has no stable time-crystaline solution.

Short-range interactions between the atoms qualitatively modify $g^r$ and lead to two separatrices in Fig. 1(b). The expression for the separatrix $U_{c2}(\Delta)$, drawn with a solid line is given in Eq. (A.19), while the separatrix $U_{c1}(\Delta)$ between the white and the blue region, is defined by the condition that the energy cost of a Bogoliubov excitation, $\omega_a$, equals $\Delta$. When $U > U_{c1}$ the self-interactions of the critical mode become finite and repulsive as $\omega_c < \omega_a$, leading to a finite cavity component of the critical mode.

It is further remarkable that the sign of the self-interactions can be changed via $U$. Indeed, within the blue region in Fig. 1(b), that is, for $U_{c1} < U < U_{c2}$, the self-interactions of the critical mode are attractive: $g_r < 0$. This is due to the fact the short-range repulsion $U$, which penalizes density modulations and in particular excitation of the recoil component $\psi_{\pm Q}$, is not sufficient to counteract the decrease of energy due to coupling to the negatively detuned cavity mode. The resulting instability of the OSR solution corresponds to a subcritical Hopf bifurcation [30] of Eq. (6). On the other hand, when $U > U_{c2}$ (green region in the figure), the short-range repulsion penalizes density-modulations enough to change the sign of the self-interaction of the critical mode and thus stabilize the OSR phase. This corresponds to a transition from a subcritical to a supercritical Hopf bifurcation.

# 4 Energy redistribution

Up to this point, the OSR time crystal is found to exist in a stable fashion as a supercritical Hopf bifurcation. Its infinite lifetime is a consequence of the coherent scattering between the atomic modes with momentum $q = 0$ and $q = \pm nQ$, with $n$ being a positive integer. Only the $n = 1$ modes belong to the CM but the higher-order integer modes only lead to a small renormalization of the amplitude of the OSR phase. This happens because the $nQ$ modes all scatter effectively with the cavity such that a steady state is found with an occupation of the $n > 1$ modes that is exponentially decreasing with $n$. For this reason we refer to the manifold with all the $nQ$ modes as the extended center manifold. However, the atomic interactions allow for scattering of occupation into modes where $q \neq nQ$. We will refer to those as the not-center-manifold (NCM) modes. As these modes are not coupled effectively to the CM modes by the cavity they can exhaust the occupation in the $nQ$ modes and thereby destroy the coherent nature of the OSR phase. To fully assess the stability of the OSR phase, one has to allow for the redistribution of energy including the NCM modes. Hence, one needs to treat the many-body problem of scattering between quasi-particles and a time-dependent coherent field.

Let us first predict which NCM modes initially participate in the scattering process, assuming we are only slightly into the OSR phase. If one also considers the regime of small $U$, which is the experimentally relevant one [21], then the occupation of an NCM mode is described by a collision that is first order in $U$. If all NCM modes are initially unoccupied, then the incoming modes must belong to the center manifold. From Eqs. (5) and (6) it is seen that there are essentially five occupied components in the stable OSR phase. The dominant component is a zero frequency and zero momentum component with weight $b_0$. Then there are components carrying momentum $\pm Q$ at the frequency $\omega_c$ with weight $c_{\pm}$ and finally the smallest contribution comes from the zero momentum modes with frequency $2\omega_c$ which are weighted by $b_+$. As the scattering is momentum and energy conserving the fastest-growing NCM mode results from scattering between the atomic components $b_0$ and $c_{\pm}$ of the center manifold, as

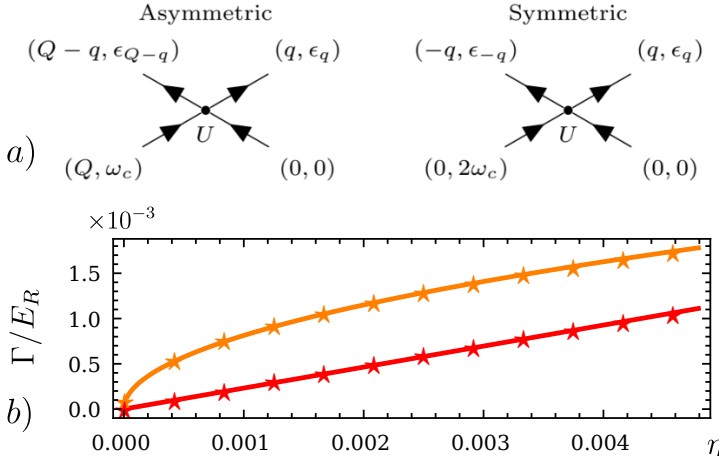

Figure 2: a) The dynamic nature of the OSR phases combined with finite atom-interaction leads to occupation of atom modes out of the center manifold, through the symmetric and asymmetric process illustrated here. b) The scaling of the growth rates, computed from the Floquet quasi-energies of the linearized equations, for the asymmetric channel marked with orange stars, with a square-root fit (orange line) and the scaling of symmetric channel marked with red stars, with a linear fit (red line). The same parameters as in Fig. 3 have been used.

illustrated in Fig. 2(a). For this process, the outgoing NCM modes with energies $\epsilon_q, \epsilon_{q'}$ have to satisfy $q + q' = Q$, $\epsilon_q + \epsilon_{q'} = \omega_c$. Since here $q \neq -q'$, we call this the asymmetric channel. Near the critical point, we can approximate $\epsilon_q$ with the Bogoliubov dispersion of the BEC excitations in the absence of the cavity field, which for small $U$ reads $\omega_B(k) \approx E_R k^2 + U$. This yields $q = Q/2 + \sqrt{\omega_c - E_R/2 - 2U}/\sqrt{2}$ and $q' = Q - q$. From the approximation of the CM components in Eqs. (5) and (6), we predict an exponential growth of these two Bogoliubov modes with a rate $\Gamma = \Gamma_{asym} \propto U\sqrt{\eta}$. This asymmetric channel can be closed off if $\omega_c < E_R/2 + 2U$, which leaves us with a different channel where the component $b_0$ scatters with $b_+$, or $c_+$ with $c_-$. Both these processes produce a symmetric NCM pair with $q = -q' = \sqrt{\omega_c - U}$. One representative process of this symmetric channel is shown in Fig. 2(a). In contrast to the asymmetric counterpart, we predict $\Gamma = \Gamma_{sym} \propto U\eta$ for the symmetric scattering processes.

In order to further verify the above predictions, we have linearized Eqs. (1) and (2) around the OSR phase and extracted the rate by computing the Floquet quasi-energies. The details of these calculations can be found in Appendix B and the resulting growth rates are shown in the lower panel of Fig. 3. It is seen that the predicted momentum (orange marks for the asymmetric channel and red mark for the symmetric channel) is only reliable close to the phase transition as the dispersion of the NCM mode is quickly modified due to the growing oscillating density modulation. We also find an additional momentum component that grows (marked in green), which arises from the scattering between a negative momentum NCM mode in the symmetric channel and the recoil component of the center manifold. The computed growth rates for the symmetric and asymmetric modes are shown in Fig. 2(b), and in both cases, an excellent agreement with our simple predictions based on Fig. 2(a) is demonstrated. Finally, in order to fully confirm our predictions, we performed a full numerical integration using a Runge-Kutta-4 routine, starting from the OSR phase at $\eta = 0.06$, corresponding to the white dashed line in the lower panel of Fig. 3. After evolving the system for 200 periods we compared the momentum distribution with the predictions based on the Floquet quasi-energies and find excellent agreement, as shown in the upper panel of Fig. 3.

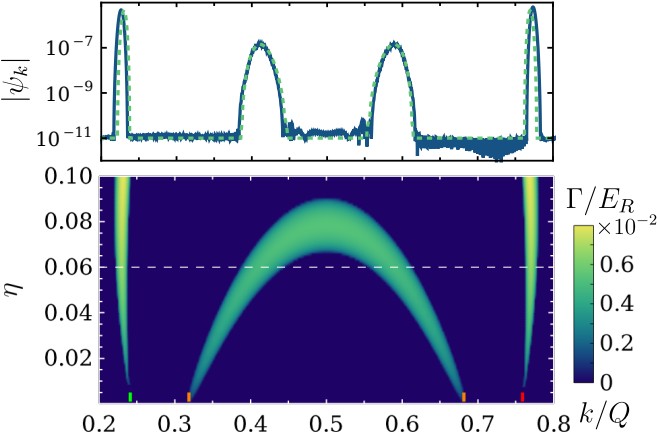

Figure 3: The lower plot shows the exponential growth rates of the atomic modes outside of the center manifold. The parameters are equivalent to those in Fig. 1 with $\Delta = 0.6E_R$ resulting in $\omega_c = 0.586E_R$, and we choose $U = 0.01E_R$. The orange ticks indicate the predicted momentum based on the asymmetric channel, while the red tick signifies the symmetric channel momentum. The green tick is the atom mode coupled to the symmetric channel through the cavity. The upper plot shows the resulting atom distribution after 200 periods at the dashed line in the lower plot, both with numerical integration of Eqs. (1) and (2) in blue and from the linearized prediction with the dashed green line.

As long as the NCM modes with the largest occupation are still small compared to unity, i.e. the time crystal has not melted yet, the CM modes still satisfy

$$\sum_{q=0,\pm Q} \left| \psi_q \right|^2 \approx 1 \, .$$

The rate of the fastest-growing NCM modes therefore sets the time scale on which the cascade becomes relevant. This time scale is explicitly given by $\tau = \ln a_0 / \Gamma$, where $a_0$ is the initial seed in the NCM modes. This initial seed is determined by the temperature of the system and energy of the fastest growing NCM mode.

## 5  Metastable nature of time crystal

In the previous section, we have established that atom interactions lead to scattering from the CM to NCM modes with $q \neq nQ$, with $n \in Z^+$. We have identified the fastest-growing NCM modes and that they grow exponentially. As these NCM modes are not coupled via the cavity to the CM modes, one expects them to evolve incoherently, such that the coherent nature of the OSR will be destroyed once the occupation of the NCM modes becomes non-negligible. To verify this, in this section, we will analyze the full numerical solution of Eqs. (1) and (2) in more detail.

Before discussing this we note that the numerical calculations require very dense momentum grids. This is because the NCM modes have to be numerically broadened such that they can be sampled on a finite grid. The artificial linewidth corresponds to a time scale, which has to be much smaller than the simulated time. Here we use a broadening of $2 \times 10^{-6}E_R$, allowing us to reach simulation times of order $10^5/E_R$. As quantum fluctuations have been neglected a small finite seed is needed for all the NCM modes and we initialize all the NCM

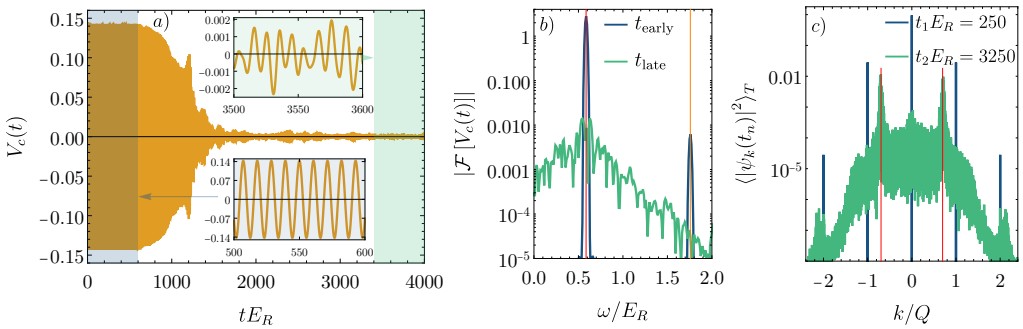

Figure 4: a) The effective cavity potential felt by the atoms resulting from the full numerical solution of Eqs. (1) and (2) for the same parameters as Fig. 3 apart from $U = 0.1E_R$. The insets show a zoomed-in view of $V_c(t)$ from the early and late time regions respectively (the arrows indicate the regions the insets are parts of). b) The power spectrum of the Fourier transform performed on the finite interval marked by the blue ($t_{\text{early}} \in \{0, 600\}E_R^{-1}$) and green ($t_{\text{late}} \in \{3400, 4000\}E_R^{-1}$) shaded regions in plot (a). The red line indicates the predicted $\omega_c$ while the orange line is at $3\omega_c$. c) The corresponding momentum distribution of the atomic modes within the stable OSR region in blue ($t_1 = 250E_R^{-1}$) and after its destruction in green ($t_2 = 3250E_R^{-1}$). The red lines indicate the momentum of the fastest-growing modes predicted by symmetric channel scattering predictions.

modes with a constant magnitude of $10^{-11}$, corresponding to $a_0$ discussed at the end of the previous section. Additionally, it is also necessary to go to large momentum values to ensure the results are fully converged (we truncate at $|k_{\text{max}}| = 5.4Q$). The resulting numerical momentum grids are thus very long $\mathcal{O}(4 \times 10^4)$. To fully understand the stability of the OSR phase it is also necessary to include the fact that a finite number of atoms gives a stochastic nature to the equations due to the openness of the cavity. This limits the magnitude of the numerical time step as the noise appears correlated on a time scale of $\Delta t$, with $\Delta t$ being the numerical time step. We therefore require that $\Delta t \ll 1/E_{\text{max}}$ where $E_{\text{max}} \sim k_{\text{max}}^2$ is the largest energy scale in our simulation. The simulations therefore require a small time step (we use $\Delta t = 5 \times 10^{-4}$ and a Runge-Kutta-2 routine for the deterministic part [31]). All parameters apart from $U$ will be kept the same as those used in Fig. 3 such that $\omega_c = 0.586E_R$ leading to a characteristic periode of $T = 2\pi/\omega_c = 10.735E_R^{-1}$. To verify that the asymmetric channel is closed when $\omega_c < E_R/2 + 2U$, while simultaneously decreasing the computation time, we consider $U = 0.1E_R$. As the symmetric channel grows with a rate proportional to $\eta$, the increase of $U$ by a factor of 10 ultimately results in an increase of the NCM growth rate by $\approx 2.45$. Consequently, the necessary simulation time is more than halved even though the asymmetric scattering channel is closed.

The resulting effective cavity potential felt by atoms, defined as

$$V_c(t) = \eta_c \frac{\eta + 1}{\sqrt{2}} \sum_{j=1,2} \text{Re}\left(\phi_j(t)\right), \tag{9}$$

is plotted in Fig. 4(a). $V_c(t)$ is good observable as it not only describes the optical potential felt by the atoms but also, up to a constant prefactor, corresponds to measuring the x-quadrature of the cavity field. For early times one observes perfectly coherent oscillations expected from the stable OSR phase. After the blue shaded region, which stops at $t = 600E_R^{-1} \approx 56T$ the amplitude of $V_c$ starts being perturbed as the occupation of the NCM modes becomes relevant. The exponential growth and following cascade lead to a sharp decrease in the amplitude of optical potential and a loss of its temporal coherence.

The loss of coherence is more clearly seen in Fig. 4(b) where the power spectrum of the Fourier transform of the two shaded regions in Fig. 4(a) is shown. For early time the spectrum is clearly dominated by a peak at the predicted frequency (marked in red) and a small contribution from a higher harmonic at $3\omega_c$. The finite width for early times is a consequence of the finite interval of the Fourier transform combined with the use of a window function to avoid aliasing. The spectrum of the long-time region clearly shows that no coherent oscillation is present in the cavity fields. At late times, there is a competition between synchronization via the cavity that favors the OSR phase, and decoherence caused by scattering involving NCM modes. The amplitude of the incoherent oscillations in this region is therefore related to the magnitude of $U$. To highlight the fact that the loss of coherence is an effect of having lost occupation in the CM modes, Fig. 4(c) shows the occupation of the atomic modes, averaged over one period $T$, for a time within the early and late time regions respectively. Here the early time occupation shows all occupation is basically in the CM modes ($q = \{0, \pm Q\}$). For late times this occupation has been completely redistributed to a continuum of momentum modes with the most occupied mode being the symmetric modes predicted at $q = \pm\sqrt{\omega_c - U}$ which are marked with red lines. It is worth pointing out that, as predicted, the asymmetric momentum modes seen in Fig. 3 are absent. The broad nature of the atomic distribution at late times is a result of the exponential growth of the initial NCM modes together with the fact that the occupied NCM modes act as sources for occupying new NCM modes, leading to an exponentially growing cascade of modes out of CM and into the NCM.

From the perspective of the growing NCM modes, this cascade can be understood as the unstable regime of parametric resonance with the center manifold modes acting as parametric drive via the short-range interactions. This is analogous to preheating in the early universe [32, 33], where the weakly interacting and oscillating, coherent inflation field decays into a cascade of exponentially growing fluctuations, leading to extreme non-equilibrium conditions inaccessible to perturbative methods and finally to thermalization [34]. Explicitly, the CM modes take the role of the inflaton field and the growing NCM modes resemble the particles created by the oscillating field. The analytic discussion we have presented corresponds to the linearized classical regime [35], which at later times will be superseded by increasingly non-linear effects leading to a cascade of even more quickly growing fluctuations that eventually thermalize [36]. At this point, a conceptual difference between the closed evolution considered for the early universe, and the driven open cavity model we discuss here emerges. Our model never fully thermalizes. Instead, it reaches a highly non-trivial state that corresponds to a compromise between thermalization due to short-ranged interactions and the flow equilibrium imposed by drive and dissipation.

## 5.1 Finite size of atom clouds

Any experiment will necessarily be performed with a finite number of atoms, $N$, in the cloud. As mentioned, in Section 2, the openness of the cavity then gives rise to a stochastic term in the cavity equations such that they take the form

$$i\partial_t \phi_j = (\Delta_j - i\kappa)\phi_j + \frac{\tilde{\eta}}{2\sqrt{2}}\sum_{k=-\infty}^{\infty}\bar{\psi}_k(\psi_{k+Q} + \psi_{k-Q}) + \sqrt{\frac{\kappa}{N}}\Delta W(t), \qquad (10)$$

where $\Delta W(t)$ is the Wiener increment [31]. To verify that the metastable OSR phase is not just robust towards atomic perturbation but also cavity noise we have performed the numerical calculations with different atom numbers (effectively changing the noise level). In principle one should compute several trajectories. Here, however, we are not interested in the noise-averaged state at late times but merely in the characteristic lifetime of the OSR phase. As the latter is much longer than any microscopic time scale of the system, it is faithfully obtained

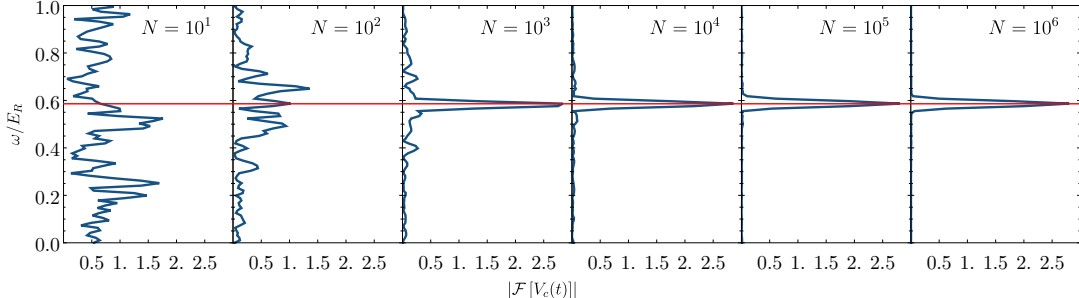

Figure 5: Plot of the Fourier spectrum of $V_c(t)$ for different atom numbers which is inversely related to the noise level. The parameters are kept equivalent to those in Fig. 4. As the time crystalline phase is only metastable the Fourier transform is performed on finite region with $t \in \{0, 600 E_R^{-1}\}$ as the early time result in Fig. 4(b). The red line indicates the predicted $\omega_c$.

already from a single trajectory. To this extent, the Fourier spectrum for different levels of noise has been plotted in Fig. 5. Here it is seen that the time-crystalline order emerges once the cloud contains $10^3$ atoms or more. It should be pointed out that we are considering a setup with a subrecoil linewidth of $\kappa = 0.4 E_R$ which is comparable to relevant experimental setups [21] in which $N = 10^5$. We thus predict that the metastable region of the OSR phase is stable against realistic noise levels.

## 5.2 Cavity parameters

Of particular experimental relevance is the question regarding the dependence of the lifetime of the OSR phase on the choice of cavity parameters. So far, we have investigated the dependence of the lifetime on the frequency of the CM components. However, the same frequency can be generated with an infinite number of different cavity parameters. This is because one can depart from the simple fully symmetric cases discussed here and include different loss rates for the different cavity modes and different coupling to the atomic cloud. Experimentally the latter can be easily controlled by changing the power in the corresponding pump sideband. So the question then becomes, given that the CM oscillates at a set frequency does it matter how the cavity is configured to achieve this frequency? Our analysis shows that, for small $U$, the scattering out of the OSR phase, which sets the lifetime, is contained in the first-order scattering processes between the atomic CM components of the atoms illustrated in Fig. 2(a). The magnitude of the atomic CM components therefore have to depend on cavity parameters directly (and not just through $\omega_c$) to impact the lifetime of the OSR phase. Using the full analytic description of the OSR phase derived in Appendix A.2, one finds that the time-averaged magnitude of the recoil components in CM, given in Eq. (A.27) is only a function of $\eta$, $U$, $\omega_c$ and $E_R$. In this model the lifetime of the OSR phase is thus independent of the specific cavity configuration. This should be contrasted with the previous results for the frequency of the OSR phase in Eq. (3), which was shown to be independent of $U$ and as such stable against interatomic interactions. Apart from this intriguing duality, the fact that the lifetime is not explicitly depending on the cavity configuration gives a large amount of freedom for experimental realizations.

# 6 Conclusion

We have analyzed the role of short-range interactions on the nature and stability of continuous time crystals in dissipative many-body systems of ultracold bosonic atoms in cavities.

First, we have shown that short-range interatomic interactions can alter the nature of the time crystal by transforming the underlying classical bifurcation from sub- to supercritical.

Second, we have studied the effect of interatomic interactions on the time crystal. Our analysis shows that when the system can only dissipate energy through the cavity, the interatomic interactions makes the time crystal inherently metastable. This can be understood from the fact that the time crystalline phase requires superradiance and thus coherent scattering of cavity photons between the atomic states. The interatomic interactions deplete the superradiant mode by populating atomic states that scatter incoherently. In the absence of an ordering principle, like a low temperature in thermal equilibrium, the depletion proceeds until the time crystalline phase ceases to exist.

We showed that near the transition, the amplitude of atomic component of the time crystalline phase is independent of the details of the underlying critical polaritonic mode. Instead it only depends on the frequency of the oscillations and the proper dimensionless distance from the critical point. The cavity dissipation is therefore not able to efficiently cool the system [37–39] (NCM modes can be de-excited only at higher order in our expansion, see Appendix B). This suggests that the metastable nature we identified is generic for these cavity systems [40], as long as the cavity linewidth is comparable to the recoil energy. We note that time-dependent Hartree-Fock approximations would miss this heating [41], as they lack collisions, and thus redistribution [42].

While we that the time crystalline phase is only metastable, we have shown that its lifetime increases significantly by considering $\omega_c < E_R/2 + 2U$ which prohibits the asymmetric scattering processes that lead to much higher growth rates. As interatomic interactions can be weak in dilute systems, the time-crystalline phases can appear stable on a time scale of the order of hundreds of periods, consistent with numerical predictions in related models [43,44]. We numerically show this statement to hold true, also in the presence of realistic noise levels.

Finally, we pointed out that cascading effect responsible for the destruction of the time crystalline phase is analogous to preheating in the early universe. It will be interesting to pursue this analogy deeper into the highly excited regime using appropriate atom-photon diagrammatic approaches [45,46].

## Acknowledgments

CHJ would like to thank Johnathan Dubois for many helpful and insightful discussions.

## A  Center manifold coefficients

The two-cavity mode system considered is a simplification of the $N$ cavity mode system discussed in [26]. In this reference the standard methods of stability analysis and bifurcation theory [29,30] is applied to understand the origin of the limit cycle, which we use as the foundation for our exploration. We refer the interested reader to reference [26] for a more detailed derivation of the critical frequency and coupling. In this appendix, we instead focus on a self-contained derivation of the analytical description of the center manifold dynamics.

The starting point is the equations of motion in Eqs. (1) and (2), which we use to define the autonomous system of non-linear first order ODE's

$$\dot{\mathbf{v}} = F(\mathbf{v}). \tag{A.1}$$

Here $\mathbf{v}$ is a vector containing the two complex cavity fields and all the complex atom fields with the different discretized momenta. As both cavity modes transfer the same longitudinal momentum ($Q$) and the BEC is initially homogeneous, the emerging critical mode only contains the cavity fields, the homogeneous atom state and the $\pm Q$ atom modes. These modes constitute the center manifold

$$\mathbf{v}_{cm} = \left(\phi_0, \phi_1, \psi_0, \psi_Q, \psi_{-Q}\right)^T. \tag{A.2}$$

The normal phase $\mathbf{X}_0 = (0,0,1,0,0)^T$ is a fixed point of $F$. When $\tilde{\eta} < \eta_c$ this fixed point is stable but becomes unstable for $\tilde{\eta} \geq \eta_c$.

As stated in the main text, slightly past the critical point the symmetry-broken state can be approximated as

$$\mathbf{u} = \sqrt{\mu} R \mathbf{v}^R e^{i\omega_c t} + c.c., \tag{A.3}$$

where $\mu = \tilde{\eta} - \eta_c$ is the absolute distance to the critical point and $\omega_c$ is the frequency of the unstable right eigenvector $\mathbf{v}^R$. We will write the equations of motion in terms of the real ($x_\alpha$) and imaginary part ($p_\alpha$) of the complex fields in which the center manifold is spanned by vectors of the form

$$\mathbf{v}^R = \left(\mathbf{v}^R_{c_1}, \mathbf{v}^R_{c_2}, \mathbf{v}^R_Q, \mathbf{v}^R_{-Q}\right)^T, \tag{A.4}$$

with $\mathbf{v}^R_\alpha = (x_\alpha, p_\alpha)^T$. The approximation in Eq. (A.3) only describes the new fixed point well if the bifurcation is of the supercritical form, which means that the self-interaction of the critical mode is repulsive. The linear coefficient in the amplitude equation Eq. (7) is given by the real part of

$$\lambda = \sum_{i,j} \mathbf{v}^L_i \left(\frac{\partial L}{\partial \mu}\right)_{i,j} \mathbf{v}^R_j, \tag{A.5}$$

where the latin indices run over all components in the center manifold $i, j \in \{x_{c1}, p_{c1}, x_{c2}, p_{c2}, x_Q, p_Q, x_{-Q}, p_{-Q}\}$. The Jacobian matrix $L = \nabla F|_{\mathbf{X}_0}$ evaluated at the normal-phase fixed point $\mathbf{X}_0$ and $\mathbf{v}^L$ ($\mathbf{v}^R$) is the left (right) critical eigenvector. We define the linear coefficient as $\gamma = \text{Re}(\lambda)$. The cubic coefficient of Eq. (7) is given by the real part of

$$g = -\frac{1}{2} \sum_{i,j,k,q} \mathbf{v}^L_i \left.\frac{\partial^3 F_i}{\partial \mathbf{X}^j \partial \mathbf{X}^k \partial \mathbf{X}^q}\right|_{\mathbf{X}_0, \mu=0} \mathbf{v}^R_j \mathbf{v}^R_k \bar{\mathbf{v}}^R_q = -\mathbf{v}^L_i (N_0)^{j,k,q}_i \mathbf{v}^R_j \mathbf{v}^R_k \bar{\mathbf{v}}^R_q, \tag{A.6}$$

where the cubic coefficient in the main text is $g^r = \text{Re}(g)$. Within the center manifold there is no contribution to quadratic contribution from $\partial^2 F$ because the center manifold obeys a reflection symmetry, which originates from the fact that the coupled Eqs. (1) and (2) posses a $\mathcal{Z}_2$-symmetry. This symmetry is that the equations are invariant under the simultaneous phase shift of the atoms by $\psi_k \to e^{i\pi k/Q}\psi_k$ and the cavity fields $\phi_j \to -\phi_j$.

## A.1 The critical eigenvector

To compute $\lambda$ and $g$ we use that the symmetry results in the right and left eigenvector components for each mode $\alpha \in \{c_1, c_2, Q, -Q\}$ are related by

$$\mathbf{v}^L_\alpha = \pm \sigma_x \mathbf{v}^R_\alpha, \tag{A.7}$$

where $\sigma_x$ is the first Pauli spin-1/2 matrix. Furthermore, the eigenvectors are normalized such that Eq. (A.7) is realized with the upper sign. The two effective interaction parameters can now be written solely in terms of the right eigenvectors

$$
\begin{aligned}
\lambda &= \sum_{i,j} \left( \left( \mathbb{1}_4 \otimes \sigma_x \right) \mathbf{v}^R \right)_i \left( \frac{\partial L}{\partial \mu} \right)_{i,j} \mathbf{v}_j^R , \\
g &= - \sum_{i,j,k,k,q} \left( \left( \mathbb{1}_4 \otimes \sigma_x \right) \mathbf{v}^R \right)_i (N_0)_i^{j,k,q} \mathbf{v}_j^R \mathbf{v}_k^R \bar{\mathbf{v}}_q^R .
\end{aligned}
\tag{A.8}
$$

The cavity eigenvector components are connected to the atomic eigenvector components through the definition of the critical eigenvector

$$
(L_0 - i\omega_c)\mathbf{v}^R = \mathbf{0} ,
\tag{A.9}
$$

which leads to the relation

$$
\mathbf{v}_{c_j}^R = -\sqrt{2}\eta_c \beta_j \begin{pmatrix} 1 & 0 & 1 & 0 \\ \frac{\kappa + i\omega_c}{\Delta_j} & 0 & \frac{\kappa + i\omega_c}{\Delta_j} & 0 \end{pmatrix} \begin{pmatrix} x_Q \\ p_Q \\ x_{-Q} \\ p_{-Q} \end{pmatrix} ,
\tag{A.10}
$$

where

$$
\beta_j = \frac{\Delta_j}{2} \frac{\Delta_j^2 + \kappa^2 - \omega_c^2 - 2i\omega_c\kappa}{\Delta_j^4 + 2\Delta_j^2 \left( \kappa^2 - \omega_c^2 \right) + \left( \kappa^2 + \omega_c^2 \right)^2} .
\tag{A.11}
$$

From the critical eigenvalue condition $\det(L_0 - I\omega_c) = 0$ one finds

$$
\begin{aligned}
\eta_c^2 \beta &= \eta_c^2 \sum_{j=1,2} \mathrm{Re}\beta_j = \frac{\omega_a^2 - \omega_c^2}{2E_R} , \\
\mathrm{Im}\beta_j &= 0 .
\end{aligned}
\tag{A.12}
$$

Linearizing around the normal phase means that the short-range interaction only couples the modes $Q$ and $-Q$ in a symmetric manner. As the cavity also couples identically to these two modes, the components of the critical eigenvector obeys $x_Q = x_{-Q} = x_a$ and $p_Q = p_{-Q} = p_a$. Using this symmetry $x_a$ and $p_a$ can be connected through Eq. (A.9) and one finds

$$
p_a = \frac{i\omega_c}{E_R} x_a = i\tilde{\omega} x_a ,
\tag{A.13}
$$

where the dimensionless frequency $\tilde{\omega} = \omega_c/E_R$ has been introduced for later convenience. Now $\mathbf{v}^R$ can be fully expressed through the parameters of our theory and $x_a$. A closed-form expression for $x_a$ can be found through the normalization condition

$$
\sum_i \mathbf{v}_i^L \mathbf{v}_i^R = 1 \rightarrow x_a = \frac{1}{2} \left( 4\eta_c^2 \sum_j \left[ \beta_j^2 \frac{\kappa + i\omega_c}{\Delta_j} \right] + \frac{i\omega_c}{E_R} \right)^{-1/2} .
\tag{A.14}
$$

This form of $x_a$ guarantees the upper sign in Eq. (A.7).

## A.2 Computing $g^r$ and $\gamma$

By substituting the critical eigenvector into Eq. (A.5) one finds the expression

$$
\begin{aligned}
\lambda &= \frac{8}{\eta_c} x_a^2 \eta_c^2 \beta = \frac{2\eta_c^2 \beta}{\eta_c \left( 4\eta_c^2 \sum_j \left[ \beta_j^2 \frac{\kappa + i\omega_c}{\Delta_j} \right] + \frac{i\omega_c}{E_R} \right)} \\
&= \frac{\omega_a^2 - \omega_c^2}{E_R} \frac{1}{\eta_c \left( 4\eta_c^2 \sum_j \left[ \beta_j^2 \frac{\kappa + i\omega_c}{\Delta_j} \right] + \frac{i\omega_c}{E_R} \right)} .
\end{aligned}
\tag{A.15}
$$

The expression for $g$ is

$$g = x_a^2 |x_a|^2 E^R \left( \tilde{U} \left( 3 + 2\tilde{\omega}^2 + 3\tilde{\omega}^4 \right) + 4 \left( 1 - \tilde{\omega}^2 \right) \left( 3 + \tilde{\omega}^2 \right) \right)$$

$$= x_a^2 |x_a|^2 E_R W_a \left( \tilde{U}, \tilde{\omega} \right), \tag{A.16}$$

where the dimensionless interaction is defined as $\tilde{U} = U/E_R$.

It is clear that the only part that makes both $\lambda$ and $g$ complex is in $x_a^2$. As the coefficients for our theory are related to the real part of $\lambda$ and $g$, it is relevant to extract the real part of $x_a^2$

$$\text{Re}\left(x_a^2\right) = \eta_c^2 \kappa \frac{\sum_j \frac{\beta_j^2}{\Delta_j}}{\left| \left( 4\eta_c^2 \sum_j \left[ \beta_j^2 \frac{\kappa + i\omega_c}{\Delta_j} \right] + \frac{i\omega_c}{E_R} \right) \right|^2}$$

$$= \frac{\eta_c^2 \kappa}{2 \left| \left( 4\eta_c^2 \sum_j \left[ \beta_j^2 \frac{\kappa + i\omega_c}{\Delta_j} \right] + \frac{i\omega_c}{E_R} \right) \right|^2} \sum_j \Delta_j \frac{\left( \Delta_j^2 + \kappa^2 - \omega_c^2 \right)^2 + 4\omega_c^2 \left( \Delta_j^2 - \omega_c^2 \right)}{\left( \Delta_j^4 + 2\Delta_j^2 \left( \kappa^2 - \omega_c^2 \right) + \left( \kappa^2 + \omega_c^2 \right)^2 \right)^2}. \tag{A.17}$$

This directly shows that the only dependence on $U$ in $x_a$ is through $\eta_c$ in Eq. (4). Due to the complexity of the full closed form expression of $g$ it is insightful to consider the behavior of $\text{Re}\left(x_a^2\right)$ and $W_a$ separately.

First considering $W_a$

$$W_a \left( \tilde{U}, \tilde{\omega} \right) = \tilde{U} \left( 3 + 2\tilde{\omega}^2 + 3\tilde{\omega}^4 \right) + 4 \left( 1 - \tilde{\omega}^2 \right) \left( 3 + \tilde{\omega}^2 \right). \tag{A.18}$$

The interesting feature of $W_a$ is the fact that it has a sign change through a zero-crossing at a critical frequency $\tilde{\omega}_0$ such that $W_a \left( \tilde{U}, \tilde{\omega}_0 \right) = 0$. The closed form expression for $\tilde{\omega}_0$ is

$$\tilde{\omega}_0 = \sqrt{\frac{\tilde{U} - 4 + 2\sqrt{2}\sqrt{8 - 4\tilde{U} - \tilde{U}^2}}{4 - 3\tilde{U}}} = \sqrt{1 + \frac{\tilde{U}}{2} + \mathcal{O}(\tilde{U}^2)}. \tag{A.19}$$

This exactly defines the separatrix $U_{c2}$ shown as a black line in Fig. 1(b). Fig. 1(b) is plotted as a function of $\Delta$ and not $\omega_c$ because of the atom instability. In the regime where $\kappa < E_R$, the equations simplify because near $\Delta \sim E_R$ one has that $\omega_c \approx \Delta$. This means that one can replace $\tilde{\omega}_0$ with $\Delta/E_R$ instead of substituting in the full expression in Eq. (3). The relevant quantity that one should compare $\tilde{\omega}_0$ to is the dimensionless frequency of the bare atomic instability, which happens at

$$\tilde{\omega}_a = \sqrt{1 + 2\tilde{U}}, \tag{A.20}$$

and which sets the dashed separatrix $U_{c1}$ in Fig. 1(b). For $\tilde{U} = 0$ the frequencies $\tilde{\omega}_0$ and $\tilde{\omega}_a$ coincide, which means that there will be no cubic interactions for the atomic instability without short-range interactions. As $\tilde{U}$ is made finite we see from the expansion in Eq. (A.19) that $\tilde{\omega}_a > \tilde{\omega}_0$ for small $\tilde{U} < 1$. By keeping the full expression for $\tilde{\omega}_0$, one finds that the critical $\tilde{U}_c$ where $\tilde{\omega}_a = \tilde{\omega}_0$ is

$$\tilde{U}_c = \sqrt{2}, \tag{A.21}$$

which is the intersection point of the separatrices at finite $U$ with $\Delta = \sqrt{1 + 2\sqrt{2}} E_R$. Below this interaction strength, $\tilde{\omega}_0$ is smaller than $\tilde{\omega}_a$. The effect is that $W_a(\tilde{U}, \tilde{\omega}_a) < 0$ for all $\tilde{U} < \tilde{U}_c$.

To determine the nature of the interactions one has to determine the sign of $\text{Re}\left(x_a^2\right)$. This sign is fixed by the numerator of Eq. (A.17) and using the parametrization discussed in the main text one finds

$$\sum_{j=1,2} \Delta_j \left( \left( \Delta_j^2 + \kappa^2 - \omega_c^2 \right)^2 + 4\omega_c^2 \left( \Delta_j^2 - \omega_c^2 \right) \right) = \frac{\delta}{2} \left( \frac{\left( \delta^2 + 4\kappa^2 \right)^2}{16} + \Delta^2 \left( 3\delta^2 + 4\kappa^2 \right) + 8\Delta^4 \right). \tag{A.22}$$

So for any values of $\kappa$, $\Delta$ and $|\delta|$, the sign of $\text{Re}\left(x_a^2\right)$ is set by the sign of $\delta$. This means that for a chosen sign of $\delta$ the sign of $W_a$ determines whether the non-linear self-interactions are repulsive or attractive. Additionally this also means that $\gamma > 0$. If $\delta > 0$ then $\omega_c < \omega_a$ and both $\eta_c^2\beta$ in Eq. (A.12) and $\text{Re}\left(x_a^2\right)$ are greater than zero. If $\delta < 0$ then $\omega_c > \omega_a$ and both $\eta_c^2\beta$ and $\text{Re}\left(x_a^2\right)$ are negative such that $\gamma$ is again positive.

Next consider the fraction $\gamma/g^r$ which determines the magnitude of the stable time-crystalline phase. By using the above derived relations one can show that it scales with $\sqrt{\epsilon}$ in the limit where $\omega_c^2 \to \omega_a^2 - \epsilon$ with $\epsilon \ll \{E_R, \Delta_{1/2}, \kappa\}$

$$
\begin{aligned}
\lim_{\omega_c^2 \to \omega_a^2 - \epsilon} \frac{\gamma}{g^r} &= \lim_{\omega_c^2 \to \omega_a^2 - \epsilon} \frac{8}{\eta_c} \frac{\omega_a^2 - \omega_c^2}{2E_R} \frac{\text{Re}\left(x_a^2\right)}{\text{Re}\left(x_a^2\right)|x_a|^2 E_R W_a\left(U/E_R, \omega/E_R\right)} \\
&= \lim_{\omega_c \to \omega_a} \frac{32}{\eta_c} \frac{\omega_a^2 - \omega_c^2}{2E_R^2 W_a\left(U/E_R, \omega/E_R\right)} \left|4\eta_c^2 \sum_j \left[\beta_j^2 \frac{\kappa + i\omega_c}{\Delta_j}\right] + \frac{i\omega_c}{E_R}\right| \\
&= \lim_{\omega_c^2 \to \omega_a^2 - \epsilon} \frac{32}{\sqrt{E_R \sum_{j=1,2} \frac{\Delta_j\left(\Delta_j^2 + \kappa^2 - \omega_c^2\right)}{\omega_c^4 + 2\omega_c^2\left(\kappa^2 - \Delta_j^2\right) + \left(\Delta_j^2 + \kappa^2\right)^2}}} \frac{\omega_a^2 - \omega_c^2}{2E_R^2 W_a\left(U/E_R, \omega/E_R\right)} \\
&\qquad \times \left|4\eta_c^2 \sum_j \left[\beta_j^2 \frac{\kappa + i\omega_c}{\Delta_j}\right] + \frac{i\omega_c}{E_R}\right| \\
&= 32\sqrt{\epsilon} \frac{\sqrt{E_R \sum_{j=1,2} \frac{\Delta_j\left(\Delta_j^2 + \kappa^2 - \omega_a^2\right)}{\omega_a^4 + 2\omega_a^2\left(\kappa^2 - \Delta_j^2\right) + \left(\Delta_j^2 + \kappa^2\right)^2}}}{2E_R^2 W_a\left(U/E_R, \omega/E_R\right)} \frac{\omega_a}{E_R} + \mathcal{O}\left(\epsilon^{3/2}\right).
\end{aligned}
\tag{A.23}
$$

This is important as it proves that the AI region does not possess a stable time-crystalline solution, to leading order in $\mu$, as $\epsilon \to 0$ for the AI.

The fact that we have analytical expressions for all the important quantities also allows us to show some intriguing features of the time-crystalline phase within the center manifold. The first important feature was discussed in the main text, namely that the frequency of OSR phase is independent of the atom parameters. The second important feature we will show now is that the time-averaged occupation in the recoil field is only indirectly depending on the cavity parameters. As stated in the conclusions, this is means that our heating discussion is more generic, as it does not depend on the specific cavity configuration. If we write Eq. (5) using $x_a$ and $p_a$ the occupation in the recoil mode is given by

$$
\begin{aligned}
\left|\psi_Q(t)\right|^2 &= \frac{1}{2}\left(|x_a|^2 + |p_a|^2\right) \\
&= \frac{\mu}{4}R^2\Big(2|x_a|^2 + x_a^2 \exp(i2\omega_c t) + \bar{x}_a^2 \exp(-i2\omega_c t) \\
&\qquad + 2|p_a|^2 + p_a^2 \exp(i2\omega_c t) + \bar{p}_a^2 \exp(-i2\omega_c t)\Big).
\end{aligned}
\tag{A.24}
$$

Due to the periodicity of the system the time average is given by

$$
\left\langle|\psi_Q|^2\right\rangle_T = \int_0^{2\pi/\omega_c} \left|\psi_Q(t)\right|^2 \mathrm{d}t = \frac{\mu}{2}R^2\left(|x_a|^2 + |p_a|^2\right) = \frac{\mu}{2}R^2|x_a|^2\left(1 + \tilde{\omega}^2\right),
\tag{A.25}
$$

where $p_a$ have been eliminated through Eq. (A.13). Using the results from Eqs. (A.15) and (A.16) we find

$$R^2 = \frac{\text{Re}(\lambda)}{\text{Re}(g)} = \frac{4\left(\tilde{\omega}_a^2 - \tilde{\omega}^2\right)}{\eta_c |x_a|^2 W_a\left(\tilde{U}, \tilde{\omega}\right)}. \tag{A.26}$$

Inserting this into Eq. (A.25) we find

$$\left\langle \left|\psi_Q\right|^2 \right\rangle_T = \frac{2\eta\left(\tilde{\omega}_a^2 - \tilde{\omega}^2\right)\left(1 + \tilde{\omega}^2\right)}{W_a\left(\tilde{U}, \tilde{\omega}\right)}, \tag{A.27}$$

which only depends on the atom parameters, the OSR frequency, and the relative depth into the OSR phase, $\eta$. The same $\tilde{\omega}$ can be generated with many different cavity configurations, for example by changing $\delta$ and having a small $\kappa$ or even more generally by departing from the fully symmetric case presented here.

## B   Including fluctuations outside the center manifold

While our theory within the center manifold predicts that the time crystal is stable also with finite interactions $U$, it does not capture atom modes outside of the center manifold. The contact interaction allows occupation in the center manifold to scatter to the other atom modes with momenta different from $\pm Q$ and 0. Inside the OSR phase the NCM modes can be occupied due to the presence of the OSR. This leads to heating and potentially also the destruction of the time crystal in the long time limit. One way to describe this is to linearize around the OSR solution $\mathbf{v}_{\text{osr}}(t)$

$$\mathbf{v}(t) = \mathbf{v}_{\text{osr}}(t) + \delta\mathbf{v}(t). \tag{B.1}$$

This leads to an equation for the fluctuations

$$\begin{aligned}
\dot{\mathbf{v}} = \dot{\mathbf{v}}_{\text{osr}} + \dot{\delta\mathbf{v}} = F(\mathbf{v}_{\text{osr}} + \delta\mathbf{v}) = F(\mathbf{v}_{\text{osr}}) + \nabla F|_{\mathbf{v}=\mathbf{v}_{\text{osr}}} \delta\mathbf{v} + \mathcal{O}\left(\delta\mathbf{v}^2\right) \\
\rightarrow \dot{\delta\mathbf{v}} = \nabla F|_{\mathbf{v}=\mathbf{v}_{\text{osr}}} \delta\mathbf{v} + \mathcal{O}\left(\delta\mathbf{v}^2\right) \approx J_{\text{osr}}(t)\delta\mathbf{v},
\end{aligned} \tag{B.2}$$

where $J_{\text{osr}}(t)$ is a time-dependent matrix-valued function. Using the approximate fixed point from the analytical OSR solution we can derive an approximate form of $J_{\text{osr}}(t)$. Due to the periodicity of the OSR solution $J_{\text{osr}}(t)$ can be expanded in a discrete Fourier series of the form

$$J_{\text{osr}}(t) = \sum_{n=-4}^{4} M_n e^{in\omega_c t}. \tag{B.3}$$

The coupling to the cavity in Eq. (1) is proportional to a product of a NCM mode and a cavity field. To first order in fluctuations there is therefore no coupling between cavity fluctuations and the NCM modes. For the leading-order heating mechanism $\delta\mathbf{v}$ only includes the atom modes with momentum $k \notin \{0, Q, -Q\}$ and is therefore solely described by Eq. (1) with the cavity fields replaced by the OSR solution. It is for this reason that we are able to use the simple scattering description, discussed in the main text, to predict the momentum of the growing modes.

Because the cavity loss has already been used to stabilize the OSR phase within the center manifold this means that the cavity is not able to cool down the NCM modes at the linear level. As one includes higher orders in fluctuations the cavity fluctuations can potentially start cooling down the NCM but as this is a higher-order effect, fine tuning would be needed to make it overcome the first-order heating before the system has thermalized and the OSR phase is destroyed.

To verify our simple scattering predictions we derive $J_{\text{osr}}(t)$ from Eq. (1). The linearized equation for the NCM mode with momentum $k$ is

$$i\partial_t\psi_k = \left(-\dot{\Omega} + k^2 - U|\hat{\psi}_0|^2 + 2U\left(|\hat{\psi}_0|^2 + |\hat{\psi}_Q|^2 + |\hat{\psi}_{-Q}|^2\right)\right)\psi_k + U\left(\hat{\psi}_0^2 + 2\hat{\psi}_{-Q}\psi_Q\right)\bar{\hat{\psi}}_{-k}$$
$$+ \frac{\tilde{\eta}}{\sqrt{2}}\sum_j \text{Re}\left(\hat{\psi}_j\right)\left(\psi_{k+Q} + \psi_{k-Q}\right), \tag{B.4}$$

where the hat has been used to identify the OSR components that are approximated as unchanged within the linearization. The overall phase of the atoms is set by $\dot{\Omega}$ and chosen such that $\text{Im}\{\psi_0\} = 0$ within the center manifold [26]. Within the linearization the value is

$$\dot{\Omega} = \frac{\tilde{\eta}}{\sqrt{2}}\sum_j \text{Re}(\phi_j)\frac{\psi_Q + \bar{\psi}_Q + \psi_{-Q} + \bar{\psi}_{-Q}}{2\psi_0}$$
$$+ U\left(2 - \psi_0^2 + \frac{1}{2}\left[\left(\psi_Q + \bar{\psi}_Q\right)\left(\psi_{-Q} + \bar{\psi}_{-Q}\right) + \left(\psi_Q - \bar{\psi}_Q\right)\left(\psi_{-Q} - \bar{\psi}_{-Q}\right)\right]\right). \tag{B.5}$$

From Eq. (B.4) we see that the finite occupation of the cavity field leads to coupling of the $k$ NCM mode with the NCM mode at $k \pm Q$. As the occupation of the NCM fields are small and we consider $\omega_c < E_R$, one can truncate after one recoil kick such that $|k| < Q$. This is confirmed by the full numerical solution of Eqs. (1) and (2) shown in the main text. With this truncation each NCM, $\psi_k$, couples to the seven other fields $\{\bar{\psi}_k, \psi_{-k}, \bar{\psi}_{-k}, \psi_{k-Q}, \bar{\psi}_{k-Q}, \psi_{-k+Q}, \bar{\psi}_{-k+Q}\}$. For each value of $k$ we therefore find a $J_{\text{osr}}(t)$ given by

$$J_{\text{osr}}(t) = i\begin{pmatrix} -m_k & 0 & 0 & -g_k & -m_Q & -g_Q & 0 & 0 \\ 0 & \bar{m}_k & \bar{g}_k & 0 & \bar{g}_Q & \bar{m}_Q & 0 & 0 \\ 0 & -g_k & -m_k & 0 & 0 & 0 & -m_Q & -g_Q \\ \bar{g}_k & 0 & 0 & \bar{m}_k & 0 & 0 & \bar{g}_Q & \bar{m}_Q \\ -m_Q & -g_Q & 0 & 0 & -m_{k-Q} & 0 & 0 & -g_k \\ \bar{g}_Q & \bar{m}_Q & 0 & 0 & 0 & \bar{m}_{k-Q} & \bar{g}_k & 0 \\ 0 & 0 & -m_Q & -g_Q & 0 & -g_k & -m_{k-Q} & 0 \\ 0 & 0 & \bar{g}_Q & \bar{m}_Q & \bar{g}_k & 0 & 0 & \bar{m}_{k-Q} \end{pmatrix}, \tag{B.6}$$

with the vector $\delta\mathbf{v}^T = (\psi_k, \bar{\psi}_k, \psi_{-k}, \bar{\psi}_{-k}, \psi_{k-Q}, \bar{\psi}_{k-Q}, \psi_{-k+Q}, \bar{\psi}_{-k+Q})^T$ and the five different entries being

$$m_k = k^2 + U - 2U\left(\hat{\psi}_Q^2 + \bar{\hat{\psi}}_Q^2\right) - \frac{\tilde{\eta}}{\sqrt{2}}\sum_j \text{Re}(\hat{\phi}_j)\frac{\hat{\psi}_Q + \bar{\hat{\psi}}_Q}{\hat{\psi}_0},$$

$$m_{k-Q} = m_{k\to k-Q},$$
$$m_Q = \frac{\tilde{\eta}}{\sqrt{2}}\sum_j \text{Re}(\hat{\phi}_j) + 2U\hat{\psi}_0\left(\bar{\hat{\psi}}_Q + \hat{\psi}_Q\right), \tag{B.7}$$
$$g_k = U\left(2\hat{\psi}_Q^2 + \hat{\psi}_0^2\right),$$
$$g_Q = 2U\hat{\psi}_Q\hat{\psi}_0.$$

Inserting the OSR solutions into Eq. (B.7) one finds an analytical expression for $J_{\text{osr}}(t)$ which is periodic such that $J_{\text{osr}}(t) = J_{\text{osr}}(t + T)$ with $T = 2\pi/\omega_c$. We then employ standard Floquet theory by numerically time-evolving the eight equations over one period $T$. This allows us to find the fundamental matrix $\Phi(t)$ which is defined as the solution to

$$\partial_t\Phi(t) = J_{\text{osr}}(t)\Phi(t), \tag{B.8}$$

with the initial condition $\Phi(0) = \mathbb{1}_8$. The eigenvalues $\lambda_i$ of the monodromy matrix $M = \Phi(T)$ determines the growth rates of the NCM modes $\Gamma_i = \text{Re}(\log(\lambda_i)/T)$. To understand the initial heating effects we only need to investigate the eigenmode with the largest growth rate $\Gamma = \max(\Gamma_i)$. By Computing $\Gamma$ as a function of $k$ we are able to compute the growth rates of the different channels as plotted in Fig. 3.

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
