# Peer review of "The role of atomic interactions in cavity-induced continuous time crystals"

_SciPost Physics, doi:SciPost Phys. 17, 089 (2024)_

## Round 1 · Referee Report · Anonymous (Referee 1) · 2024-1-23

Report

In this paper the authors consider the dynamics of a BEC, described by Gross-Pitaevski mean-field equations, coupled to two coherent cavity fields. The authors show that in this setup continuous time-crystalline phases emerge and analyze the role of short-range atomic interactions. They show that the atomic interactions change the type of bifurcation which underlines the oscillating phase and that the interactions induce unstable modes which might render the oscillating phase metastable.

The paper is well written and the results seem to be reliable. The topic is timely as it can be seen from the recent experimental observation of continuous time crystals in systems of ultracold atoms coupled to optical cavities. The results presented can further the interpretation of such experiments.

For these reasons I can recommend publication of this paper in SciPost Physics after the following questions and comments have been addressed.

Questions and comments:

1) It is not clear to me how the instability rates of not-center-manifold modes relate to the lifetime of the OSR phase. I understand that the NCM unstable modes are computed by a linear analysis on top of the OSR phase. However, if one considers the non-linear dynamics of the CM and NCM modes wouldn't it be possible to still obtain an oscillating phase (albeit, with probably a modified structure compared to the OSR of the linear analysis)? Or more generally, my question is what is the long time state of the system (either steady or oscillating) when all the modes are considered? As the authors have performed the numerical integration of Eqs (1)-(2) to confirm their analytical calculations, maybe a more extended discussion of these time-dependent results would be helpful, both at short and long times.

2) In Fig. 3a the scale of psi_k for the discussed modes is very small, ~10^-6 to 10^-11, is this due to the choice of normalization, or there are other modes with a much larger contribution? In particular, what is the occupation of the mode at k=0 and the OSR mode at k=Q in this regime?

3) I would find helpful if the regime of validity of the model used would be briefly discussed, in particular which effects present in a full quantum description of the atoms-cavities setup pertinent to the conclusions might be neglected here. I understand that this point might be beyond the scope of the present work, as the authors only deal with mean-field equations for both the atoms and the cavity modes, however in the conclusions in the paragraph from line 216 the authors make a rather generic conclusion regarding the energy redistribution in an open system. For example in Bezvershenko et al., PRL. 127, 173606 (2021), quantum fluctuations beyond the mean field treatment of the atoms-cavity coupling were shown to induce an effective heating or cooling of the atomic subsystem. These effects might be important on longer timescales than considered in the present work, but could be relevant to establishing the metastable character of the states considered.

4) All 3 solutions for the critical modes presented in lines 87-89 are labeled with omega_c, this makes a bit hard to follow the subsequent discussion of Eq (4) and the paragraph afterwards.

5) Similarly for the rates labeled with Gamma in the paragraph 164-179.

6) In line 212 an equation from Appendix A is mentioned. As this equation comes after the result of several pages of calculations I recommend that the sentence in the main text to be rephrased such it is self-contained and only the appropriate Appendix section is mentioned.

7) In Appendix A, line 238 the word "thesis" should probably be replaced with "appendix" or "section"

---

## Round 1 · Referee Report · Anonymous (Referee 2) · 2024-2-5

Report

In this paper, Johansen and coauthors investigate the properties of dissipative time crystals in the presence of local interactions. The general conclusion is that the interaction could change the nature of the critical behavior, and furthermore eventually leads to the complete meltdown of the (dissipative) time crystal. The conclusions are quite interesting, and the analysis appears to be sound although I have not checked every equation explicitly. That said, the technical nature of the paper and the ambiguous presentation of some points make it hard to fully understand some key conceptual points of the paper. Here is a list of questions/ comments:

- Dissipative time crystals vs limit cycles: this is a general question not about the paper, but about the field of dissipative time crystals. What is the distinction between a dissipative time crystal and a limit cycle? A few words on this point puts this paper in a broader context.
- Are there known examples of (dissipative) limit cycles which are stable against local perturbations? If yes, what’s different about these models vs the present model?
- Remarks about inflation appear only in the abstract and then only very briefly in the conclusions. If the authors like to make a meaningful analogy, they need to further expand on this point.
- In Eq. (1), the mean-field Gross-Pitaevskii equation is used without noise. How does noise affect the general conclusions of the paper?
- The conclusions up to (including) Section 3 are believable. However, Section 4 requires some expanding and elaborating. Specifically, why does the exponential increase of k modes (other than 0, +Q, -Q) indicate instability? In a regime where the collective modes show (time crystalline) ordering, it would be quite natural for the other k modes to “piggyback” on these modes. That is, I would expect that in the same phase that the collective modes are nonzero or oscillating, other k modes follow suit as well. In fact, the authors mention only briefly at the end of Appendix B that “As one includes higher orders in fluctuations the cavity fluctuations can potentially start cooling down the NCM but as this is a higher-order effect, fine tuning would be needed to make it overcome the first-order heating before the system has thermalized and the OSR phase is destroyed.” I am not convinced that any fine tuning is needed. In fact, symmetry allows other k modes to find a nonzero expectation value or to exhibit persistent oscillations, and the exponential growth identified by the authors could mean that they just rise to saturate to their ordered values.
- A comment on the presentation: I appreciate the authors’ effort to make the paper more accessible by relegating the technical parts to the appendices, and focus on the main physics in the main text. However, parts of the paper are hard to parse, as one must go back and forth in the text. Furthermore, the technical methodology used in the appendices are not sufficiently explained, and a bit too technical for a non-expert reader to penetrate. I’d suggest adding simple explanations/background for any technical tool that’s used in the paper.

---

## Round 2 · Referee Report · Anonymous (Referee 1) · 2024-7-7

Report

I think the authors have satisfactorily addressed the issues raised in my previous report. The new section on the numerical solution of the full equations is a good addition to the message of the paper and puts the analytical results in a better context. Thus, I can recommend the publication of the paper in its present form.

Recommendation

Publish (meets expectations and criteria for this Journal)

---

## Round 2 · Author Response

We have implemented the referees comment and in particular this lead to adding a new section focusing on the numerical solution of the full equations. Here it was additional shown how the previous conclusions are stable against realistic noise levels.

---

## Round 2 · List of Changes

1) Added sentence about stability towards noise in abstract. Line: 17-19 2) Explained structure of paper. Line: 59-70
3) Additional paragraph discussing validity of model. Line: 100-109 4) Skecthed linear expansion method: 119-122 5) Clarified different superradiant phases. Line: 122-131 + 150-151 6) Made AI critieria more transparent. Line: 135-138 7) Changed title of section 4 to: "Energy redistribution" 8) Expanded upon the the effect of NCM modes vs. CM modes. Line: 201-213 9) Added paragraph to clarify the nature of the energy redistrubution mechanism and how it is approximated. Line: 216-226 10) Included new variables to better highlight the difference between the assymetric and symmetric scattering channels. Line: 232 + 237 11) Fixed typo in $\omega_c$ inequality. Line: 233 12) Added paragraph highlighting the small occupation of the NCM modes in the numerical results presented section 4. Line: 255-260 13) Added new section 5 titled "Metastable nature of time crystal". This section highlights the metastable nature of the TC through a careful numerical investigation and shows that the results are also stable towards noise. Line: 261-375 14) New paragraph in the conclusion highlight phenomonolgy behind our results. Line: 382-405 15) Clarified initial reference in Appendix A. Line: 413-417

---

## Editorial Decision

published